# Sentiment Analysis and Emotion Recognition from Speech Using Universal Speech Representations

**DOI:** 10.3390/s22176369

**Published:** 2022-08-24

**Authors:** Bagus Tris Atmaja, Akira Sasou

**Affiliations:** National Institute of Advanced Industrial Science and Technology, Tsukuba 305-8560, Japan

**Keywords:** affective computing, sentiment analysis, speech emotion recognition, sentiment analysis and emotion recognition, universal speech representation

## Abstract

The study of understanding sentiment and emotion in speech is a challenging task in human multimodal language. However, in certain cases, such as telephone calls, only audio data can be obtained. In this study, we independently evaluated sentiment analysis and emotion recognition from speech using recent self-supervised learning models—specifically, universal speech representations with speaker-aware pre-training models. Three different sizes of universal models were evaluated for three sentiment tasks and an emotion task. The evaluation revealed that the best results were obtained with two classes of sentiment analysis, based on both weighted and unweighted accuracy scores (81% and 73%). This binary classification with unimodal acoustic analysis also performed competitively compared to previous methods which used multimodal fusion. The models failed to make accurate predictionsin an emotion recognition task and in sentiment analysis tasks with higher numbers of classes. The unbalanced property of the datasets may also have contributed to the performance degradations observed in the six-class emotion, three-class sentiment, and seven-class sentiment tasks.

## 1. Introduction

Humans communicate not only by exchanging linguistic information with others but also through paralinguistic and non-linguistic information [1]. Para-linguistic information includes intention, attitude, and style (which are not inferable directly from linguistics). Non-linguistic information includes sentiment and emotion. Sentiment can be regarded as a sub-category of emotion. It is close to (or the same as) valence in dimensional emotion. Nevertheless, previous studies of sentiment and categorical emotion have been performed independently.

An attempt to include sentiment information in the task of speech emotion recognition was undertaken in [2]. In that study, the authors accommodated the sentiment-aware method to improve speech emotion recognition performance by training a dataset with the automatic speech recognition (ASR) loss and cross-entropy sentiment loss functions. The model was then fine-tuned using the concordance correlation coefficient loss function (CCCL) to predict valence, arousal, and dominance. CCCL has been proven to lead to better scores than other loss functions [3]. The results showed an improvement in valence prediction in speech emotion recognition, highlighting the connection between sentiment analysis and valence prediction.

As in human–human communication, machines (computers) should be able to recognize non-linguistic information in human–machine communication, particularly in sentiment analysis and emotion recognition (SAER). This ability is important for more natural human–machine interactions. Suppose that a robot is able to detect the affective states of humans. In that case, a robot will also be able to show empathy and sympathy to humans by acting, speaking, or showing different reactions.

SAER entails the use of artificial intelligence (AI) to extract sentiment and emotion information from modalities such as text, audio, and video [4]. Although there is no consensus in distinguishing between the two, the majority of researchers view sentiment as part of emotion, as explained previously. This view is backed by an argument that sentiment analysis uses a simplified binary or three-class categorization, whereas emotion recognition relies on a deeper analysis of affective states and sensitivities. However, differentiation matters when these terms are used in marketing, e.g., in feedback from customers, responses may include statements such as “I hate your product” (emotion) or “Your service is bad” (sentiment). Nevertheless, the similarity between the two can be used as a starting point to evaluate the use of AI models in sentiment analysis (SA) and emotion recognition (ER) tasks.

Instead of studying sentiment analysis and categorical emotion recognition separately, we performed an integrated study of sentiment analysis and emotion recognition from speech using the same model trained on a sentiment and emotional speech corpus. This method enabled the evaluation of a model employed in different tasks, instead of a single task. On the other hand, we were able to track which task gained the most benefit from the model by using different variants of the model with different evaluation metrics.

Research on spoken sentiment analysis is derived from natural language processing (NLP). In NLP, the main object of research is text; hence, in early spoken sentiment analysis research, speech was transcribed into text to analyze the sentiments of transcriptions (e.g., [5,6]). Nowadays, the advancements in deep learning have shortened the paths for extracting sentiment information from speech by bypassing transcription and learning directly from acoustic features [7,8]. The use of acoustic features instead of text is advantageous because speech is richer in affective information than plain text. Other researchers have fused multimodal information (video, text, speech) to improve the performance of speech sentiment analysis [9,10].

Research on speech emotion recognition is a step ahead of spoken sentiment analysis. One of the reasons and indicators for this argument is the availability of the speech emotion recognition (SER) corpus, which contains a greater quantity of datasets than the available speech or spoken sentiment analysis corpus. For a comparison, Zadeh et al. [11] listed nine speech sentiment analysis datasets and 13 SER datasets. Furthermore, Atmaja et al. [12] listed 19 SER datasets for acoustic-linguistic emotion recognition.

In this study, we aimed to contribute to affective computing research by evaluating three variants of universal speech representations for three sentiment analysis tasks and an emotion recognition task independently. Since sentiment and emotion are close (sentiment is often represented as valence, which is an emotional attribute), it is reasonable to evaluate the use of the same model for both problems. This case is not a multitask learning case but four single-task learning cases. The same model, except for the output layer, was built for the evaluation of four different tasks of speech sentiment analysis and emotion recognition. The input used for the models consisted of universal speech representations. Recent research on self-supervised learning has shown the benefit of using universal speech representations with speaker-aware pre-training (UniSpeech-SAT) [13] for analyses of non-linguistic information, particularly in emotion recognition tasks, over other self-supervised learning (SSL) techniques including wav2vec, HuBERT, and WavLM. By employing this recent SSL model, we expected to gain insights into the performance of the same model on different paralinguistic information extraction tasks from speech.

## 2. Related Work

Research on speech sentiment analysis and emotion recognition is not new. Previous studies have focused on building feature extraction methods or classifiers for general speech processing or specific cases. In this section, we highlight the previous research on sentiment analysis and emotion recognition from speech and emphasize the key differences in our approach in this study compared to those of the previous studies.

Bertero et al. [14] performed speech emotion and sentiment recognition for interactive dialogue systems. The authors annotated the TED-LIUM release 2 corpus for emotion recognition and employed Movie Review and Twitter corpora for sentiment analysis. Those authors utilized raw speech for emotion recognition and word2vec for sentiment analysis. It is clear that although the authors proposed to undertake both sentiment analysis and emotion recognition, they used two different models for each task (using different datasets). For emotion recognition, the authors evaluated the use of the SVM and CNN methods. For sentiment analysis, the authors evaluated the use of the CNN and LIWC (Linguistic Inquiry and Word Count) methods.

In [15], the authors proposed a method to extract features from visual and textual modalities using a deep CNN for emotion recognition and sentiment analysis. The method, which is called multiple kernel learning, is a feature selection method to combine data from different modalities effectively. The multimodal data consisted of audio, video, and text. The proposed method with feature selection slightly improved the performance of multimodal fusion without feature selection. The gap in the performance of multimodal fusion over any single modality was large, highlighting the benefit of the use of multimodal fusion over the feature selection method.

The authors of [7] trained a robust wav2vec 2.0 system for dimensional emotion recognition, including sentiment recognition. In that work, the sentiment was treated the same as valence. The authors reported significant improvements in valence prediction by fine-tuning the robust wav2vec 2.0 for a dimensional emotion recognition task. This method of acoustic-only emotion recognition closed the gap between valence and other emotion dimensions (arousal and dominance), in which the previous methods suffered from low performance in valence predictions. Before that study, SER employed linguistic information from transcriptions to improve predictions of valence.

Compared to these previous studies, in this study we focus on the evaluation of the same model for different sentiment analysis and emotion recognition tasks. The models’ performances were evaluated based on weighted and unweighted accuracies. Our goal was to find feasible tasks that were performed well and consistently by a specific model.

## 3. Methods

### 3.1. Dataset

We employed (and also motivated by the freely available) CMU-MOSEI dataset [11,16]. Although the dataset was intended for the study of expressed sentiment and emotions in multimodal language, we only utilized the audio data. The target application was audio-based applications such as telephone calls, customer services, and virtual assistants. The raw CMU-MOSEI contains 23,259 utterances (Table 1) for each task. There are three sentiment tasks and an emotion task. The sentiment tasks are divided into two-class tasks (2-c, positive and negative), three-class tasks (3-c, positive, neutral, and negative), and seven-class tasks (7-c, [−3, −2, −1, 0, 1, 2, 3]). The emotion task was derived from Ekman’s six basic emotions, consisting of happiness, sadness, anger, surprise, disgust, and fear. Note that the data reported in this paper were different in terms of the number of samples from the original reported in [16]; the original authors split the samples into a fixed length of sequences. For instance, the number of samples for fear in the original reference [16] was close to 1900, whereas the authors only obtained 452 samples.

### 3.2. UniSpeech-Sat Models

Pre-trained models were used to extract acoustic features for the input of the classifier. The models were three variants of UniSpeech-SAT [13], which were trained on Librispeech, Librivox, VoxPopuli, and Gigaspeech datasets. The Base model was trained in 400k steps on LibriSpeech 960 h audio. The Base+ and Large models were trained in 400k steps on 94k large-scale diverse data (10k Gigaspeech + 24k VoxPopuli + 60k Librivox) [13]. Both Base and Base+ have 94.68M parameters while the Large has 316.61M parameters. The models were built on top of HuBERT models [17] by integrating contrastive-loss with the self-supervised learning objective function and utilizing an utterance-mixing strategy for data augmentation. The models showed higher performance than other SSLs in cases of non-linguistic information tasks, including emotion recognition.

Figure 1 shows our method for evaluating different tasks using the same dataset (CMU-MOSEI) and similar methods. For each task, the configuration is different only in terms of the number of nodes in the output layer of the transformers. These numbers of nodes depend on the number of classes in the task, e.g., two nodes for two-class sentiment analysis and six nodes for six-class emotion recognition. Note that all tasks were performed independently in a single-task manner; the method does not involve multitask learning that predicts all four tasks simultaneously.

## 4. Experiments

We used the S3PRL toolkit [18] for experiments with our methods.The upstreams consisted of three variants of UniSpeech-SAT models: UniSpeech-SAT Base, UniSpeech-SAT Base+, and UniSpeech-SAT Large. The downstream was the ‘mosei’ task with “num_class” values of either 2, 3, 6, or 7. The configuration “num_class:6” was used for emotion recognition, whereas other configurations were used for sentiment analysis. Table 2 shows the details of the hyperparameters used in the experiment. The values of these parameters were placed in the config.yaml file inside the MOSEI downstream directory.

## 5. Results and Discussion

### 5.1. General Results

We present our evaluation of spoken sentiment analysis and emotion recognition with universal speech representations in terms of weighted and unweighted accuracies. Weighted accuracy (WA) refers to the overall accuracy determined by dividing the number of correct predictions (true positive and true negative) by the number of total predictions. The unweighted accuracy (UA) is the average accuracy per class (average recall). Table 3 shows the results of our experiments in all tasks.

Comparing the sizes of the UniSpeech-SAT models, it is clear that the large model learned better from the larger data set than the other two smaller models. The improvements from Base, Base+, and Large models were about 1%–2% for both WA and UA. Comparing each task, for the lowest number of classes, i.e., two-class sentiment analysis, we observed feasible performance (WA and UA). There are two possible reasons for these results. The number of the classes was small, and it is possible that the nature of sentiment lacks a neutral class. Although the first explanation is well established, it is necessary to confirm the second by experimenting with the same number of samples in two- and three-class spoken sentiment analysis using other datasets.

### 5.2. Per Task Evaluations

For the two-task sentiment analysis, the results can be detailed by means of a confusion matrix showing the performance of the model for each class, as shown in Figure 2. The model (UniSpeech-SAT Large) recognized positive sentiments at higher rates than negative sentiments due to the number of positive samples compared to negative samples (16,576 vs. 6683). Nevertheless, despite the low portion of negative samples (28% of total samples), the obtained performance results still show that the large model could learn from these data.

For the three-task sentiment analysis, the confusion matrix is shown in Figure 3. It can be observed that the performance in regard to negative sentiment improved (from 53% to 57%), whereas the performance in regard to positive sentiment decreased (from 93% to 87%). An interesting result is the low performance in relation to neutral sentiment (27%) with the 5100 samples. This number was included among the positive sample group in the previous two-class sentiment analysis (here, the positive sample was split into positive and neutral samples). There is a debate in psychological research as to whether neutral affect exists [19] or does not exist [20]. Our results support the second theory, as indicated by the fact that the two-class sentiment analysis gained more feasible (by WA, for practice) and reliable (by UA, for details) results than the three-class sentiment analysis with a neutral state.

The confusion matrix of the six-class emotion recognition task is shown in Figure 4. In this task, the score of WA showed moderate accuracy (64%). However, the confusion matrix shows that the (large) model failed in the recognition of emotional categories except for happiness, where it achieved 94% accuracy/recall. The reason for these failed predictions was the large size of the happiness sample; more than half of the data (14,567 samples, 63%) were labeled as happy.

The last task was a seven-class sentiment analysis, and its confusion matrix is shown in Figure 5. The WA score for this task was the worst among other tasks, whereas its UA score was only better than that of six-class emotion recognition. As shown in the confusion matrix, the recall scores for this seven-class spoken sentiment analysis only displayed the best results for sentiment classes “−2” and “1”, supporting the low UA score. The reason for this UA result is the unbalanced distribution of the samples for each class. The lowest number of samples in label/class “3” (with 675 samples in total) was mostly predicted as having the label “2” (60%) in the test set. The recall for this label “3” was the worst, with a percentage of only 0.91% among the others. In contrast, the label “1” with the most samples (7576) obtained the highest accuracy (68%); this evidence supports the previous unbalanced distribution problem.

We found in all tasks that an unbalanced distribution is a critical issue in deep-learning-based sentiment analysis and emotion recognition. The computational load in this study was another issue; we needed to perform the experiments in a very small batch of two samples to avoid memory limitation errors (out of memory, OOM). The size of the UniSpeech-SAT Large model, aside from the size of the data, was responsible for this limitation. Future research may be directed toward overcoming these issues.

### 5.3. Benchmarking with Previous Studies

To gain additional knowledge, we performed a benchmark comparison with the results of previous studies (Table 4). Although our results did not achieve the highest score, we found that when using only unimodal acoustic information, the evaluated method in this research achieved a competitive score with multimodal fusion methods in two-class sentiment analysis (81.4% vs. 82.5%). Although the score obtained in two-class sentiment analysis was competitive with others, our scores in six-class emotion recognition and seven-class sentiment analysis were remarkably lower than the multimodal fusion results. This issue should be addressed in future studies. Note although the dataset was the same (the CMU-MOSEI dataset), the number of instances used may have been different. Some references [9,21] did not provide the number of instances, whereas references [22,23] mentioned the use of 23,454 instances.

Within this study, we also compared UniSpeech-SAT with traditional MFCC features. MFCC is the most common acoustic feature used in audio processing, including neonatal bowel sound detection. In [24] the authors employed 24 coefficients of MFCC with a 25 ms window and 13 ms stride for bowel sound detection to assist in auscultation. Here, the number of MFCC coefficients was 13 with their first and second derivatives. The stride and window size were also 10 ms and 25 ms, respectively. The performance of the UniSpeech-SAT model was remarkably higher than that of MFCC in the two-class, three-class, and seven-class sentiment analysis (about 10% gap in WA). For six-class emotion recognition, the gap of 2% with UniSpeech-SAT was still higher than that of MFCC.

**Table 4 sensors-22-06369-t004:** Comparison of accuracies (WA%) of the results of previous studies using the CMU-MOSEI dataset; A: audio, T: text, V: video; scores in italics denote UA.

Method	Modality	Acc${}^{2}$	Acc${}^{3}$	Acc${}^{6}$	Acc${}^{7}$
RAVEN [9,25]	A + T + V	79.1	-	-	50.0
MCTN [9,26]	A + T + V	79.8	-	-	49.6
M.Rout [9,27]	A + T + V	81.7	-	*81.4*	51.6
MuIT [9,22]	A + T + V	82.5	-	-	51.8
M${}^{3}$ [9]	A + T + V	82.5	-	-	51.9
DCCA [23]	A + T	69.4	-	-	-
MAT [21]	A + T	82.0	-	-	-
MFCC	A	71.7	54.0	62.9	34.5
UniSpeech-SAT	A	81.4	65.3	64.9	44.8

We believe that the research presented in this paper will have a broad positive impact on the speech emotion recognition and spoken sentiment analysis communities due to the following considerations. We adopted an open science approach, utilizing open data and methods which are fully reproducible and replicable. The CMU-MOSEI dataset can be downloaded freely without the need for a prior agreement. The model and toolkit, S3PRL, are also available in an open repository. The configuration of the experiments used to obtain the results is clearly stated in this paper. Readers can replicate our experiment and make further improvements and modifications, e.g., by balancing the number of samples for each class for each task and/or evaluating other pre-trained models and comparing the results with the scores reported in this paper. The first author welcomes correspondence from readers on any detail missing from this paper.

## 6. Conclusions

In this study, we independently evaluated three tasks of sentiment analysis and a task of emotion recognition based on speech using similar methods. The methods were universal speech representations with speaker-aware pre-training models used as acoustic feature extractors and a transformer architecture used as the classifier. The evaluation of large-scale pre-trained speech embeddings (UniSpeech-SAT Large) on these affective speech tasks consistently resulted in superior performance among other variants; however, this was only feasible for two-task sentiment analysis. The other tasks were not feasible (UA < 60%), which could have been caused by several possible factors, mainly influenced by the unbalanced distribution of the data.

Although in this study we conducted sentiment analysis and categorical emotion recognition independently, future studies could merge these tasks into a multitask learning approach, predicting both sentiment and categorical emotion simultaneously. Multitask learning benefits from applying information from one task to the other tasks, e.g., predicting the naturalness of speech could improve emotion and naturalness predictions in multitask learning [28].

## Figures and Tables

**Figure 1 sensors-22-06369-f001:**
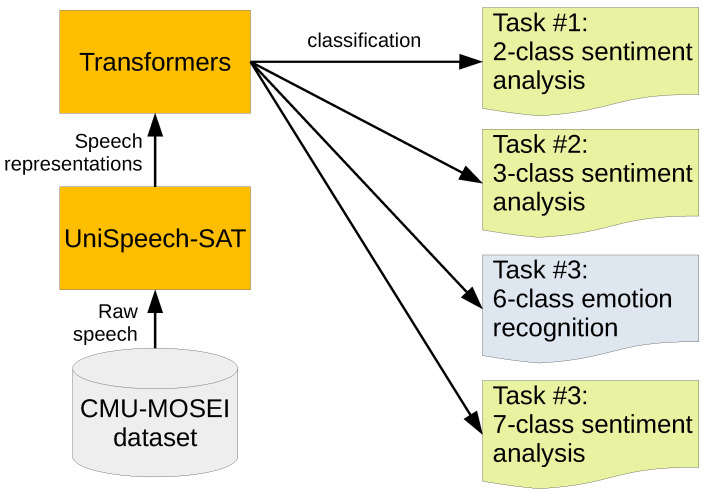
Flow diagram of the data processing method from the dataset to each classification task.

**Figure 2 sensors-22-06369-f002:**
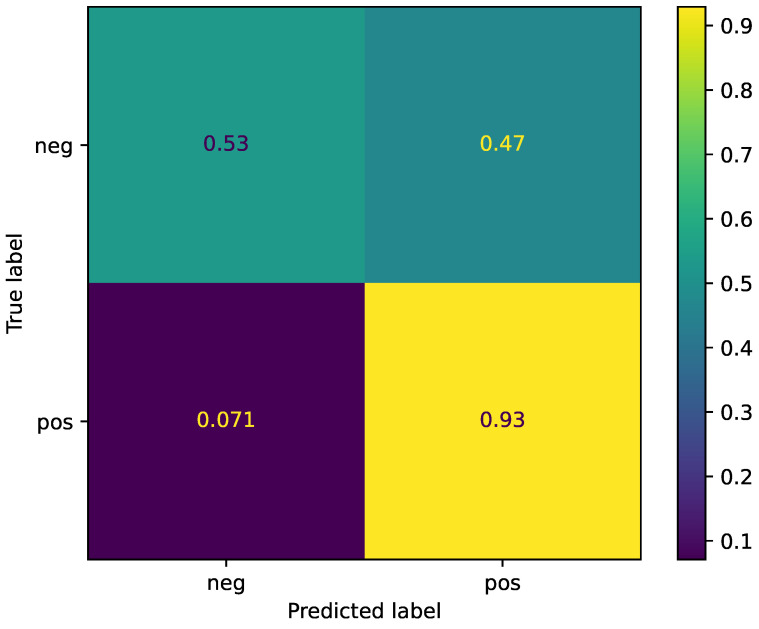
Normalized confusion matrix for two-class sentiment analysis.

**Figure 3 sensors-22-06369-f003:**
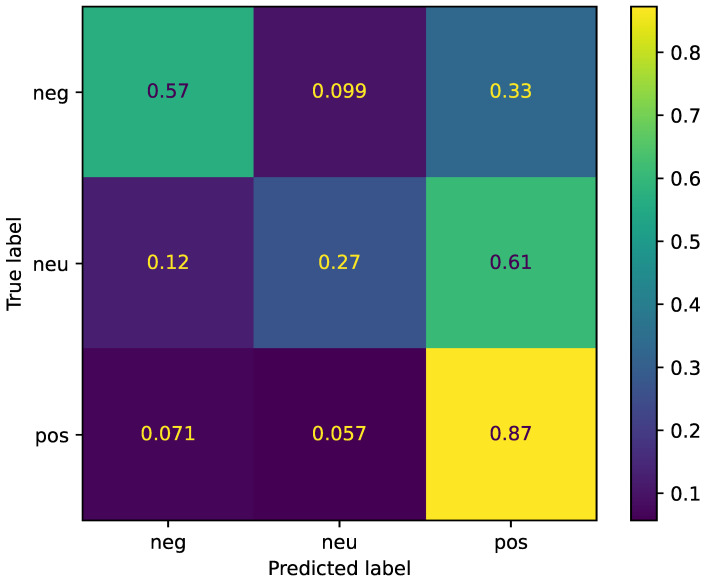
Normalized confusion matrix for three-class sentiment analysis.

**Figure 4 sensors-22-06369-f004:**
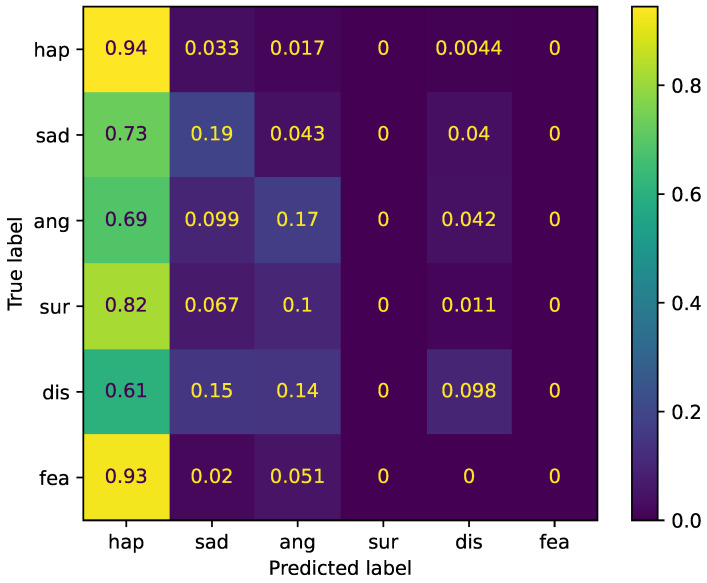
Normalized confusion matrix for six-class emotion recognition.

**Figure 5 sensors-22-06369-f005:**
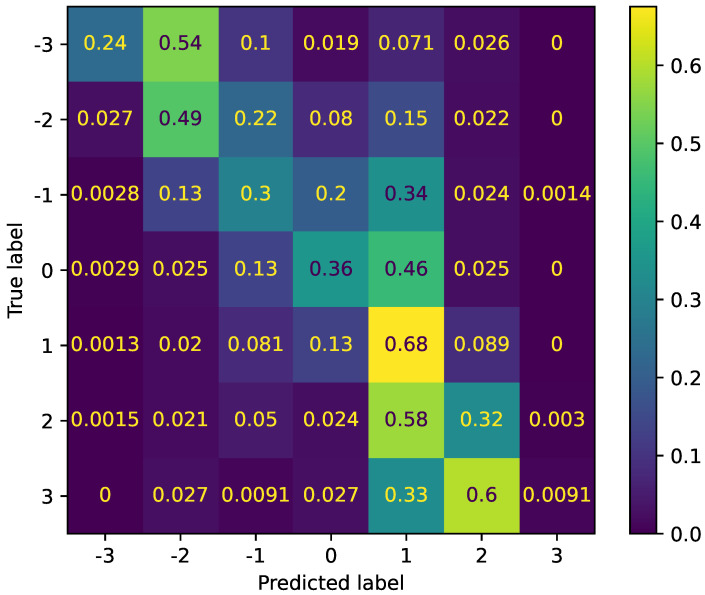
Normalized confusion matrix for seven-class sentiment analysis [−3, 3].

**Table 1 sensors-22-06369-t001:** Distribution of samples for each sentiment and emotion label for different classes (c).

Sentiment				Emotion	
**Labels**	**# 2-c**	**# 3-c**	**# 7-c**	**Labels**	**# 6-c**
−3	-	-	821	happiness	14,567
−2	-	-	2253	sadness	3782
−1	6683	6683	3609	anger	2730
0	-	5100	5100	surprise	437
1	16,576	11,476	7576	disgust	1291
2	-	-	3225	fear	452
3	-	-	675	-	-
Total	23,259				

**Table 2 sensors-22-06369-t002:** Hyperparameters used in the experiments.

Hyperparameter	Value
Optimizer	AdamW
Learning rate (LR)	0.0002
LR scheduler	Linear
Batch size	2
#Total steps	10,000
#Eval/save steps	250
#Warm up steps	1000
#Workers (CPU)	10
#GPU	4
#Transfomer dim	128
#Head	2
#Encoding layers	2

**Table 3 sensors-22-06369-t003:** Weighted and unweighted accuracies (WA and UA) for sentiment analysis and emotion recognition tasks using the MOSEI dataset; Bolds indicate the highest scores.

Task	WA	UA
	UniSpeech-SAT Base	
2-class sentiment	78.68	69.46
3-class sentiment	61.33	53.75
6-class emotion	64.33	22.01
7-class sentiment	40.63	29.12
	UniSpeech-SAT Base+	
2-class sentiment	79.36	68.85
3-class sentiment	63.06	55.14
6-class emotion	64.52	22.15
7-class sentiment	42.64	31.12
	UniSpeech-SAT Large	
2-class sentiment	**81.36**	**72.97**
3-class sentiment	**65.27**	**57.12**
6-class emotion	**64.95**	**23.39**
7-class sentiment	**44.85**	**34.20**

## Data Availability

Not applicable.

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
