# Peer review of "Sentiment Analysis and Emotion Recognition from Speech Using Universal Speech Representations"

_sensors, 2022, doi:10.3390/s22176369_

Round 1

Reviewer 1 Report

Sentiment Analysis and Emotion Recognition from Speech

Using Universal Speech Representations

---------------------

The paper carries out sentiment analysis and emotion recognition from speech using universal speech representations. The paper has potential but requires major revision.

1. Abstract: gradations or degradations?

2. Did the paper use any class balancing methods?

3. Please merge small paragraphs for readability.

4. Did the paper use exactly the same split of data for the comparison with SOTA?

5. The paper lacks recent speech representation and classification techniques. Please cite and mention how the current representation is different from the one explained in:

https://ieeexplore.ieee.org/abstract/document/9784899

This paper uses MFCCs for the audio representation.

6. Is it possible to use MFCC and classify it to compare it with the current study?

Reviewer 2 Report

Even though the problem of the paper from my point of view is its novelty (the scope of the paper is too narrow and the contribution of the paper is limited), the results are good and can be published after a minor revision.  In Keywords: "universal speech representation)"  can be changes to "universal speech representation".  The length of the paper can be shortened. For example, Figures 2-5 can be merged in one figure. 

Round 2

Reviewer 1 Report

Given that the authors addressed the concerns, the reviewer is inclined to accept.  Please proofread the manuscript.

Author Response

We thank the reviewer for the comment.

We have already proofread the manuscript.